# Genetic Parameters for Selected Traits of Inbred Lines of Maize (*Zea mays* L.)

Adrian Cyplik [1], Aleksandra Sobiech [2], Agnieszka Tomkowiak [2] and Jan Bocianowski [1,*]

[1] Department of Mathematical and Statistical Methods, Poznań University of Life Sciences, Wojska Polskiego 28, 60-637 Poznań, Poland; adrian.cyplik@up.poznan.pl

[2] Department of Genetics and Plant Breeding, Poznań University of Life Sciences, Dojazd 11, 60-632 Poznań, Poland; aleksandra.sobiech@up.poznan.pl (A.S.); agnieszka.tomkowiak@up.poznan.pl (A.T.)

[*] Correspondence: jan.bocianowski@up.poznan.pl; Tel.: +48-61-8487143

**Abstract:** This paper presents an estimation of the parameters connected with the additive (*a*) effect, additive by additive (*aa*) epistatic effect, and additive by additive by additive (*aaa*) interaction gene effect for nine quantitative traits of maize (*Zea mays* L.) inbred lines. To our knowledge, this is the first report about *aaa* interaction of maize inbred lines. An analysis was performed on 252 lines derived from Plant Breeding Smolice Ltd. (Smolice, Poland)—Plant Breeding and Acclimatization Institute-National Research Institute Group (151 lines) and Małopolska Plant Breeding Ltd. (Kobierzyce, Poland) (101 lines). The total additive effects were significant for all studied cases. Two-way and three-way significant interactions were found in most analyzed cases with a considerable impact on phenotype. Omitting the inclusion of higher-order interactions effect in quantitative genetics may result in a substantial underestimation of additive QTL effects. Expanding models with that information may also be helpful in future homozygous line crossing projects.

**Keywords:** inbred lines; maize; additive effect; epistasis; three-way epistasis

## 1. Introduction

Currently, maize, along with wheat and rice, are among the economically most essential grain species [1]. Maize is one of the most important crops that allow for the continuous growth of the world's population. The importance of maize is greatly influenced by its high production potential and the possibility of extensive use of crops (food source, industrial raw material, various types of animal feed). In recent years, the area of maize cultivation in the world ranges from 140 to 145 million hectares, while its production exceeds 600 million tons of grain. The continuous increase in the range of maize cultivation is related to the breeding progress, which includes the use of heterosis and the creation of hybrids with fewer climate requirements [2–4]. It is also crucial to have access to more and more modern breeding methods and cultivation technologies. The demand for new varieties of maize is constantly growing, which makes it the subject of intensive genetic and breeding research.

From the breeding point of view, in the case of quantitative traits, the interaction of genes in the creation of a given trait is critical. It is known that each of the genes that determine a polygenic trait is inherited according to Mendel's laws. Three independent genes and two alleles at the locus are enough for the frequency distribution of the genotypes to approach the normal distribution. However, such a situation occurs in a very simplified case, i.e., when two-allelic loci with an identical, linear phenotypic effect interact additively [5].

In fact, the phenotypic effect of different loci may be different. Loci can be multiallelic. There can be partial or full dominance. We can also deal with overdomination, couplings, and the effect of interallelic interaction. Modifier genes may also appear, modifying the functions of other genes [6,7].

The traits with minor polygenic effects of each genotype overlap with environmental effects, i.e., the variability of the polygenic feature is influenced by the environment. Consequently, it is impossible to identify the genotype based on the phenotype [8].

Therefore, one of the primary tasks of applied genetics is to explain the relationship between genotype and phenotype. Thus, the phenotypic observation of a quantitative trait includes the environmental, genetic, as well as genotype by environment interaction [9,10].

Genetic analyses of various maize genotypes conducted for years have shown that gene interactions—dominance, epistasis, and pleiotropy—play an essential role in the evolution of the maize phenotype. It has long been noticed that many of the loci that differentiate maize and teosinte are pleiotropic. The latest examination of the regulatory architecture of tb1 provides a detailed understanding of pleiotropy for a single domestication gene. The tb1 is pleiotropic across many traits such as apical dominance, length of lateral branches, growth of leaves on the lateral branches, pedicellate spikelet development, and root architecture [11,12].

Genetic analyses are of the most importance for understanding the variability of phenotypic traits. The influence on those traits by genes and their pairwise interactions are well known [1,13,14], but considering higher-order interactions is still a novelty in this kind of studies [15].

The aim of this paper was an estimation of the three parameters connected with gene effects: additive (*a*), additive by additive (*aa*) epistatic, and additive by additive by additive (*aaa*) interaction for nine quantitative traits of maize inbred lines. Acquiring this new insight, especially on the effect of the higher-order interaction, was considered for the potential homozygous line crossing. On the basis of available literature, this is the first report concerning the presence of estimation of the additive × additive × additive interaction gene effect of maize inbred lines.

## 2. Materials and Methods

### 2.1. Plant Material

A collection of 252 inbred maize lines was evaluated. A total of 151 lines were derived from Plant Breeding Smolice Ltd. Plant Breeding and Acclimatization Institute-National Research Institute Group; however, 101 lines were derived from Małopolska Plant Breeding Ltd. These lines (252) were deployed in two localities, 120 km apart, Smolice (51°42′12″ N, 17°10′10″ E) and Kobierzyce (50°58′1″ N, 16°55′50″ E).

### 2.2. Experimental Field Trails

A field experiment with 252 inbred lines was set up on plots of 10 m$^2$, in a randomized complete block design (RCBD) in three replicates, in two locations. During the experiments, observations of quantitative traits were carried out. After the harvest, in the first half of November, biometric measurements were made. We observed the following features: cob length, cob diameter, core length, core diameter, the number of rows of grain, the number of grains in a row, thousand kernel weight (TKW), and yield. Measurements were carried out on ten randomly selected cob per replicate.

The soils in Smolice were made of limestone and chalk formations. For this solid valuation III B, and wheat-good complex by quality class, and maize can yield high both on wheat-beet and rye soils, class IV b.

Maize was sown by hand. Mineral fertilization was adapted to the needs of maize grown for grain harvesting. We applied 350 kg·ha$^{-1}$ polifoska 6 and 160 kg·ha$^{-1}$ urea. The nutrient content was as follows: before winter grind: 70 kg P$_2$O$_5$·ha$^{-1}$, 21 kg N·ha$^{-1}$, 105 kg K$_2$O·ha$^{-1}$ in polifoska form, and before sowing 73.6 kg N·ha$^{-1}$ in urea form. Chemical weed control was performed with a plot sprayer.

### 2.3. Genetic Parameters

The following formula estimated the additive (*a*) gene effect:

$$\widehat{a} = \frac{1}{2}\left(\overline{L}_{max} - \overline{L}_{min}\right), \tag{1}$$

where $\overline{L}_{min}$ and $\overline{L}_{max}$ are the means for the extreme groups (minimal and maximal lines, respectively). The group of minimal (maximal) lines consists of the lines which contain, theoretically, only alleles reducing (increasing) the value of the trait [16]. Groups of extreme lines were identified by the quantile method [17] in which lines with mean values smaller (bigger) than 0.03 (0.97) quantile of the empirical distribution of means are assumed as minimal (maximal) lines.

The following formula estimated the additive × additive (*aa*) epistasis interaction effect [18,19]:

$$\widehat{aa} = \frac{1}{2}\left(\overline{L}_{max} + \overline{L}_{min}\right) - \overline{L}, \tag{2}$$

where $\overline{L}$ is the mean of all inbred lines.

The additive × additive × additive (*aaa*) three-way epistasis interaction effect was estimated by the following formula [15]:

$$\widehat{aaa} = \frac{1}{2}\left(L_{max} + L_{min}\right) - \overline{L}, \tag{3}$$

where $L_{min}$ and $L_{max}$ are the lines with minimal and maximal mean value, respectively.

The test statistics to verify hypotheses about genetic parameters different than zero are given by:

$$F_a = \frac{MS_a}{MS_e}, \; F_{aa} = \frac{MS_{aa}}{MS_e} \text{ and } F_{aaa} = \frac{MS_{aaa}}{MS_e}, \tag{4}$$

where $MS_a$, $MS_{aa}$, $MS_{aaa}$, and $MS_e$ are mean squares for parameter *a*, epistasis *aa*, epistasis *aaa*, and residual, respectively.

## 3. Results

The obtained results for estimates of the total additive (*a*) gene effect, the total additive × additive × additive (*aaa*), alongside the two-way epistasis interaction effect, were presented in Table 1. The total additive effects were significant for all studied cases. Results show that 67% (24 out of 36 cases) of the total three-way epistasis effect was statistically significant. All of the *aaa* interactions were significant for the number of rows of grain in all locations/origins. Significant two-way interactions were found in 58% of studied cases (21 out of 36), and as above, the number of rows of grain was the only one with all locations significant. Between traits, the three-way interactions estimates range was relatively low for most cases, ranging between 0.58 and 1.38. In four cases, however, those values were much higher: the number of rows of grain, the number of grains in a row, mass of grain from the cob and thousand kernel weight (estimates scopes respectively: 2.75, 7.62, 15.66, 20.67). The smallest scope of estimates was found for the cob diameter. The same situation occurred for pairwise interactions, with the same lowest and highest traits. The lowest values of the estimates range were found for cob diameter (0.30) and the highest for the number of rows of grain, a mass of grain from the cob and thousand kernel weight (2.45, 3.12, 11.59, 15.45, respectively). Only for the number of rows of grain, the influence was always positive both in *aa* and *aaa* interactions for location/origin. In other cases, influences were mixed but often paired, having the same positive or negative impact on *aa* and *aaa* interaction effect.

**Table 1.** Mean values and the total additive (*a*) effects as well as the total interaction *aa* and *aaa* effects for the 252 (151 from Smolice, 101 from Kobierzyce) inbred lines of maize estimated on the basis of only phenotypic values.

| Location | Origin | Parameter | Cob Length | Cob Diameter | Core Length | Core Diameter | The Number of Rows of Grain | The Number of Grains in A Row | Mass of Grain from the Cob | Thousand Kernel Weight | Yield |
|---|---|---|---|---|---|---|---|---|---|---|---|
| Kobierzyce | Smolice | mean | 15.55 | 3.97 | 15.44 | 2.16 | 15.73 | 27.91 | 114.8 | 265.15 | 4.59 |
| | | *a* | 4.03 *** | 0.92 *** | 4.06 *** | 0.71 *** | 4.80 *** | 6.97 *** | 47.41 *** | 91.79 *** | 1.90 *** |
| | | *aa* | 0.09 | 0.09 | 0.06 | 0.12 ** | 1.07 ** | −0.01 | −3.19 * | −1.94 | −0.13 * |
| | | *aaa* | 0.06 | 0.18 * | 0.24 | 0.19 ** | 1.27 ** | 0.42 | −4.19 * | 0.47 | −0.17 * |
| Kobierzyce | Kobierzyce | mean | 15.37 | 3.85 | 15.17 | 2.03 | 15.41 | 27.44 | 106.61 | 254.47 | 4.26 |
| | | *a* | 3.83 *** | 0.66 *** | 3.82 *** | 0.59 *** | 4.56 *** | 8.61 *** | 49.44 *** | 91.02 *** | 1.98 *** |
| | | *aa* | −0.38 * | −0.08 | −0.52 * | −0.01 | 1.59 *** | −0.05 | −1.88 | 3.02 | −0.08 |
| | | *aaa* | −0.76 * | −0.06 | −0.80 ** | 0.03 | 1.59 *** | 0.39 | 1.04 | −1.34 | 0.04 |
| Smolice | Smolice | mean | 13.36 | 4.09 | 13.6 | 2.24 | 15.01 | 24.97 | 94.38 | 288.9 | 3.78 |
| | | *a* | 3.40 *** | 0.75 *** | 3.28 *** | 0.96 *** | 4.40 *** | 11.15 *** | 42.72 *** | 103.9 *** | 1.71 *** |
| | | *aa* | 0.44 * | 0.03 | 0.42 * | −0.39 *** | 0.86 ** | −3.11 *** | 3.97 * | −4.06 | 0.16 * |
| | | *aaa* | 0.30 * | 0.24 ** | 0.57 * | −0.39 *** | 0.99 ** | −7.20 *** | 3.68 * | −3.56 | 0.15 * |
| Smolice | Kobierzyce | mean | 12.91 | 3.98 | 13.3 | 2.28 | 14.91 | 25.32 | 87.46 | 276.22 | 3.5 |
| | | *a* | 4.28 *** | 0.75 *** | 3.56 *** | 0.33 *** | 4.11 *** | 9.06 *** | 43.81 *** | 97.5 *** | 1.75 *** |
| | | *aa* | −0.47 * | −0.21 ** | −0.08 | 0.04 | 1.20 ** | −2.27 ** | −7.62 ** | 11.39 * | −0.30 ** |
| | | *aaa* | −1.08 ** | −0.37 ** | −0.30 * | 0.04 | 1.76 *** | −3.32 *** | −11.98 *** | 17.11 ** | −0.48 *** |

\* $p < 0.05$; \*\* $p < 0.01$; \*\*\* $p < 0.001$.

## 4. Discussion

Maize is a model plant, so it is an excellent research object [20]. This species, along with rice, is the most commonly grown crop for human and animal consumption. It is a species grown for herb and forage [21]. According to USDA, the world's maize production in 2019/20 was 1116 million tons, and in 2020/21 increased by 42 million tons. Climate change and the massive demand for maize, conditioned by the natural increase, lead the research on genomic regions, which are significant for agronomy [22]. The modern approaches to maize cultivation predominately crosses plants between inbred lines. These hybrids test white vigor (heterosis) and more broad agronomic properties than parents. However, the heterosis effect cannot be estimated based on the phenotype parents line because its basis is unknown. The hypothesis is that heterosis based on completed damaged homozygous alleles is hidden in one line in inbred parents [23].

Heritability, in all its complexity, always gives us an answer in the end. However, fully predicting the outcome before it happens is still out of our reach. One of the missing pieces of information may be found in higher-order genetic interaction studies [24]. The main effects of genes in maize were analyzed by Chaikam et al. [25]. However, epistasis was estimated by Mihaljevic et al. [26], Blanc et al. [27], and Stange et al. [28]. A different, heuristic approach to assessing the effects of genes has been presented by Bocianowski et al. [29].

One of the best-known applications of heritability in quantitative genetic studies of traits is its predictive role, which helps determine the reliability of phenotypic value as an indicator of breeding value. High genetic progress and high heritability scores provide the best conditions for selection [30].

Genotype can be assessed by phenotype, representing the genotypic value in the dynamic environment. Gene expression results from additivity, dominance, and epistasis, affecting the expression of quantitative traits in a population. Inheritance results from the interaction of genotype and phenotype, the variation of which is useful in the selection process [31]. In the analysis shown by Amegbor et al. [32], phenotypic and genetic correlations showed similar trends for the traits analyzed.

According to Gazala et al. [33], a combinations of genes with the diverse large of effects and modes of action (additive, dominance, and epistasis) are involved in the inheritance of complex quantitative traits such as grain yield and the features of the yield structure, including a significant non-crossover/crossover interaction with the environment [33].

Complex maize traits are defined by polygenes exhibiting additive, dominance, epistasis, and gene–environment interaction effects. An example of such traits can be the ones of the leaf. Many statistical methods such as the additive and full genetic models have been developed to discover complex plant traits. However, most genetic association studies focus on additive effects, ignoring non-additive epistasis, environment–gene interactions, and interaction effects [34]. It leads to the absence of the heritability problem, which can significantly impact phenotypic variation [35].

Ma et al. [36] presented that the values of the additive effect of the genes were greater than the non-additive effects. Plant height and ear height were easily affected by environmental factors. The broad-sense heritability and narrow-sense heritability of kernel depth are higher. They are significantly affected by non-additive effects so that they can be selected in the late generation. Similar results for 1000-kernel weight in spring barley hulled and hull-less lines obtained Bocianowski et al. [10].

Results show that 67% of the total three-way epistasis effect and 58% of two-way interactions were statistically significant in the studied cases. These outcomes highlight the importance of including such information in similar studies. High ranges of estimates can be explained by divergent experimental situations with varying locations, origins, and growing conditions of plants [17]. The type of influence, positive or negative, of both two and three-way interaction effects was often paired. This leads to a conclusion that comprehensive analysis of QTLs effects (additive, additive by additive epistatic, and additive by additive by additive interaction) is necessary in the breeding program purposes. Obtained results can be the basis for selection in breeding program. It is the most suitable

method for plant breeding applications, and its main benefit is faster progress in yield and other traits.

## 5. Conclusions

A significant additive gene action effect in a specific population means that selection beginning in the early generations gives hope for obtaining transgressive homozygous lines.

The detection of epistatic effects enables a better understanding of the interaction of individual genes and allows for a more precise estimation of the effects of individual genes.

Higher-order interactions, although commonly neglected, often occur with significant influence on phenotypic traits. Presented results show that two-way interaction and three-way interaction effects are often similar in influence type (positive or negative) and estimate value itself. Omitting the inclusion of higher-order interactions effect in quantitative genetics may result in a considerable underestimation of additive QTL effects.

Further studies of and with the inclusion of higher-order genetic interactions are necessary.

**Author Contributions:** Conceptualization, A.C. and J.B.; methodology, A.C. and J.B.; software, A.C. and J.B.; validation, A.C., A.S., A.T. and J.B.; formal analysis, A.C. and J.B.; investigation, A.C., A.S., A.T. and J.B.; resources, A.S. and A.T.; data curation, A.S. and A.T.; writing—original draft preparation, A.C., A.S., A.T. and J.B.; writing—review and editing, A.C., A.S., A.T. and J.B.; supervision, J.B.; project administration, J.B.; funding acquisition, J.B. All authors have read and agreed to the published version of the manuscript.

**Funding:** This research received no external funding.

**Institutional Review Board Statement:** Not applicable.

**Informed Consent Statement:** Not applicable.

**Data Availability Statement:** Not applicable.

**Conflicts of Interest:** The authors declare no conflict of interest.

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
