# Peer review of "Genetic Parameters for Selected Traits of Inbred Lines of Maize (Zea mays L.)"

_applsci, doi:10.3390/app12146961_

Round 1

Reviewer 1 Report

The research paper is interesting and informative, however, some minor revision is needed. Please find my comments in the manuscript.

Author Response

Response to Reviewer 1 Comments

Reviewer #1

Point 1: The research paper is interesting and informative, however, some minor revision is needed. Please find my comments in the manuscript.

Response: Thank you very much.

Review Comments

Point 2: The objective of the article Genetic Parameters for Selected Traits of Inbred Lines of Maize (Zea Mays L.) is estimation of the parameters connected with the different additive effects and genetic interactions regarding nine quantitative traits of maize (Zea mays L.) inbred lines.

Response: Thank you very much.

Point 3: A fairly large number of maize lines (252) was used, which makes the findings of this research significant for the scientific peers, as well as for the maize breeders and producers worldwide.

Response: Thank you very much.

Point 4: In the Introduction part of the manuscript, the subject was well addressed and citation of the literature is adequate and up to date.

Response: Thank you very much.

Point 5: The Material and Methods are well presented and the methods used are appropriate for this kind of research.

Response: Thank you very much.

Point 6: The Result and the Discussion part of the article tackle the subject of the research appropriately, and elaborate the findings well, by making appropriate conclusions and references to other previously conducted researches.

Response: Thank you very much.

My comments are:

Point 7: 1. Extend the Conclusion with one to two sentences emphasizing the significance of the results obtained in the research.

Response: We extended Conclusions. We added two sentences: "A significant additive gene action effect in certain population means that selection begining in the early generations gives hope for obtaining transgressive homozygous lines." and "The detection of epistatic effects enables a better understanding of the interaction of individual genes and allows for a more precise estimation of the effects of individual genes."

Point 8: 2. Use synonyms when a word is repeated within the sentence multiple times, if possible.

Response: We corrected manuscript.

Point 9: 3. I suggest the English language be checked for minor corrections.

Response: We corrected manuscript.

Reviewer 2 Report

1.  Please add more comprehensive aims and also the novelty of the study in the abstract and introduction part

2.  Several comments for the results and discussion are available in the manuscript.  A more comprehensive discussion is needed to elaborate on the great finding of the experiment

3.  Conclusion also has to be more comprehensive and aligned with the purpose of the research

Author Response

Response to Reviewer 2 Comments

Reviewer #2

Point 1: Please add more comprehensive aims and also the novelty of the study in the abstract and introduction part.

Response: We extended aims and added information about this work novelty both in the abstract and introduction.

Point 2: Several comments for the results and discussion are available in the manuscript. A more comprehensive discussion is needed to elaborate on the great finding of the experiment.

Response: We corrected manuscript.

Point 3: Conclusion also has to be more comprehensive and aligned with the purpose of the research.

Response: We extended Conclusions. We added two sentences: "A significant additive gene action effect in certain population means that selection begining in the early generations gives hope for obtaining transgressive homozygous lines." and "The detection of epistatic effects enables a better understanding of the interaction of individual genes and allows for a more precise estimation of the effects of individual genes."

Round 2

Reviewer 2 Report

The revised version is sufficient to be proceeded further